# Sustainable Kapok Fiber-Derived Carbon Microtube as Broadband Microwave Absorbing Material

**DOI:** 10.3390/ma15144845

**Published:** 2022-07-12

**Authors:** Aichun Long, Pengfei Zhao, Lusheng Liao, Rui Wang, Jinlong Tao, Jianhe Liao, Xiaoxue Liao, Yanfang Zhao

**Affiliations:** 1School of Materials Science and Engineering, Hainan University, Haikou 570228, China; longaichun1997@163.com (A.L.); 990359@hainanu.edu.cn (J.L.); xiaoxueliao@hainanu.edu.cn (X.L.); 2Key Laboratory of Tropical Crop Products Processing of Ministry of Agriculture and Rural Affairs, Agricultural Products Processing Research Institute, Chinese Academy of Tropical Agricultural Sciences, Zhanjiang 524001, China; lsliao@catas.cn (L.L.); wangruifly@163.com (R.W.); jinlongt1983@163.com (J.T.); 3Hainan Provincial Key Laboratory of Natural Rubber Processing, Zhanjiang 524001, China

**Keywords:** biomass-derived carbon, kapok fiber, electromagnetic loss, microwave-absorbing performance

## Abstract

The design of hierarchical structures from biomass has become one of the hottest subjects in the field of microwave absorption due to its low cost, vast availability and sustainability. A kapok-fiber-derived carbon microtube was prepared by facile carbonization, and the relation between the structure and properties of the carbonized kapok fiber (CKF) was systematically investigated. The hollow tubular structures afford the resulting CKF composites with excellent microwave-absorbing performance. The sample with a 30 wt.% loading of CKF in paraffin demonstrates the strongest microwave attenuation capacity, with a minimum reflection loss of −49.46 dB at 16.48 GHz and 2.3 mm, and an optimized effective absorption bandwidth of 7.12 GHz (10.64–17.76 GHz, 2.3 mm) that covers 34% of the X-band and 96% of the Ku-band. Further, more than 90% of the incident electromagnetic wave in the frequency from 4.48 GHz to 18.00 GHz can be attenuated via tuning the thickness of the CKF-based absorber. This study outlines a foundation for the development of lightweight and sustainable microwave absorbers with a high absorption capacity and broad effective absorption bandwidth.

## 1. Introduction

Electromagnetic pollution has become one of the top concerns in modern daily life due to the excessive use of wireless communication devices, high-power signal base stations and even household WIFI transmitters [1,2]. In this context, microwave-absorbing materials (MAMs) with high attenuation capacity over a wide frequency are designed to address electromagnetic radiation and interference [3,4,5]. Traditionally, metals and alloys are used as MAMs, but their heavy weight, low mechanical flexibility, poor corrosion resistance and advancements in smart electronic devices have restricted further application [6,7,8]. Alternatively, carbon materials such as carbon nanotube and graphene have been considered promising candidates due to their unique advantages, including low density, tunable electrical conductivity and good environmental stability [9,10]. It has been confirmed that carbon-based hierarchical architectures demonstrate high adsorption capacity and broad effective absorption bandwidth. The three-dimensional structure is capable of elongating the reflecting and scattering paths, thus generating new interfaces and adjusting the impedance matching [11,12]. However, expensive raw fossil materials and complicated fabrication procedures (such as chemical vapor deposition, arc discharge, solvent exfoliation, etc.) of these materials are the main obstacles to their practical application [13]. Therefore, the need to explore sustainable raw materials in order to produce carbon-based MAMs with a simple, economical, and efficient method is critical.

Sustainable biomass-derived carbonaceous materials have captured increasing attention of late, becoming one of the hottest topics of current research due to their easy availability, cost-effectiveness, high yield, renewability, eco-friendly properties and versatile fabrication [14,15,16]. Confirmed by extensive studies, the fabrication of MAMs using low-cost biomass as raw materials has proven to be a promising environmentally friendly approach [17,18]. Specifically, carbonization not only maintains the natural evolution-induced hierarchical structures, including periodic patterns and well-organized porous structure, but also generates innumerous nanopores as well as defects. This helps to decrease bulk density, improve impedance matching and boosts microwave attenuation capacity. Previously, carbon materials originated from biomass such as wood [18], spinach stem [19], walnut shell [20], mango leaf [21], bread [22] etc., and were fabricated as lightweight MAMs. In order to further improve impedance matching and attenuation capacity, biomass-derived carbon is always integrated with magnetic components in an optimal structure [23,24,25]. However, the enhanced microwave-absorbing performance of magnetic nano-block-anchored biomass carbon was achieved at the sacrifice of the native characteristics of pure biomass carbon-material-based EM absorbers such as ultralight EM absorbers, which have excellent chemical resistance. Therefore, it remains a challenge to design lightweight, biomass-derived, carbon-based MAMs with high microwave attenuation capacity over a broad frequency range.

As regular architectures in biomass, the tubular structure is ubiquitous and abundant in basal wood, catkin, polar bear hair and cancellous bone [26]. Extensive studies have confirmed that biomass-derived tubular carbon materials possess many distinctive features that are different from their counterparts that have an irregular structure [27]. In this context, extensive efforts have been made to fabricate biomass-derived carbon from tubular or hollow materials by extending their use in water treatment [28], thermal management [29], and energy conversion [30]. It has been confirmed that the presence of multiscale (from nanoscale to microscale) pores in tubular, biomass-derived carbon not only facilities impedance matching by abundant air–solid interfaces, but also contributes to microwave attenuation via multiple reflections [31]. Possessing a typical hollow structure with a thin cell wall of 0.8~1.0 μm and large lumen of more than 80% porosity, kapok fiber (KF) originates from a tree grown primarily in South Asia, and contains intrinsic properties similar to other types of tubular biomass [32]. Carbon microtube derived from KF has been developed and utilized as adsorbents for organic dyes [33], sound absorption materials [34], catalysis [30], etc. However, to the best of our knowledge there are no reports on the use of KF-based microwave-absorbing material. To this end, a sustainable carbon microtube for use as a broadband-microwave-absorbing material was prepared by pyrolyzing KF in an inert atmosphere. As expected, this demonstrated an excellent microwave-absorbing performance due to the synergistic effect from the hollow structure and good dielectric loss. This work provides a new strategy for the innovative design of biomass-derived tubular materials, with a potential application in the field of microwave attenuation.

## 2. Materials and Methods

### 2.1. Materials

Kapok fibers were collected from the South Subtropical Botanical Garden at the Chinese Academy of Tropical Agricultural Sciences (Zhanjiang, China). High-purity argon (>99.999%) was provided by the Zhanjiang Oxygen Factory (Zhanjiang, China). Absolute ethyl alcohol was purchased from the Sinopharm Chemical Reagent Co., Ltd. (Shanghai, China). All of the chemical reagents in the experiment were analytically pure, and were used without further purification. All the water involved in the experiment was distilled water that had been produced in our library with a Milli-Q reverse osmosis system.

### 2.2. Preparation of CKF

CKF were prepared by carbonizing the KF. Briefly, the collected KF was washed with distilled water and absolute ethyl alcohol to remove any impurities, and then vacuum dried at 60 °C for 6 h. Then, the as-purified KF was placed into a typical ceramic crucible and carbonized in a tube furnace under argon atmosphere, with a programmed temperature from 25 °C to 600 °C and a heating rate of 10 °C/min. After being pyrolyzed at 600 °C for 2 h and then naturally cooled to room temperature, the final black product was obtained.

### 2.3. Characterizations

**FTIR.** The chemical structure of the samples was examined using a Nexus 470 Fourier transformation infrared spectrophotometer (FTIR) (Thermo Nicolet, Waltham, MA, USA) in a wavenumber range of 400–4000 cm^−1^.

**TGA.** Thermogravimetric analysis (TGA) is performed on a STA 449C thermogravimetric analyzer (NETZSCH, Selb, Germany) at a heating rate of 10 °C/min under a nitrogen atmosphere with a flow rate of 50 mL/min.

**XPS.** The element identification and heteroatom functional group distribution were measured by X-ray photoelectron spectra (XPS) (Shimadzu, Nakagyo-ku, Japan). 

**XRD.** X-ray diffraction (XRD) patterns were recorded using a D8 Advance X-ray diffractometer (Bruker, Bremmen, Germany) with a Cu Ka radiation (λ = 1.5418 Å) at a scanning speed of 5°/min from 10° to 80°. 

**Raman.** Raman spectra were collected using a Lab RAM HR Evolution confocal Raman spectrometer (HORIBA, Longjumeau, France) in a range of 100–4000 cm^−1^. 

**BET.** Pore structures were characterized by nitrogen adsorption analyses implemented on an ASAP 2460 Brunauer–Emmett–Teller (BET) analyzer (Micromeritics, Norcross, GA, USA).

**SEM.** The morphology of all samples was visualized by a S4800 Scanning Electron Microscope (SEM) (Hitachi, Tokyo, Japan) with an accelerate voltage of 3 kV. 

**Electromagnetic parameters.** The electromagnetic parameters were determined using a N5244A vector network analyzer (Agilent, Santa Clara, CA, USA) in the frequency range of 2–18 GHz. Samples were prepared by pressing (0.5 Mpa, 1 min) the homogeneous CKFs/paraffin mixture into a toroidal-shaped pipe with an outer diameter of 7.00 mm, an inner diameter of 3.00 mm and thickness of 2.00 mm. The loadings of CKF in samples were 5 wt.%, 10 wt.%, 20 wt.%, 30 wt.% and 40 wt.%, which were designated as CKF-5, CKF-10, CKF-10, CKF-20, CKF-30 and CKF-40, respectively.

## 3. Results

### 3.1. Overview of Fabrication and Nanostructure

As a natural biomass material derived from the kapok tree (*Bombax ceiba* L., Figure 1a), mainly growing in South Asia, KF can be easily obtained from nature on a large scale and at low cost. Moreover, KF naturally features hollow microstructures of more than 80% porosity that are in favor of improving impedance matching and attenuation capacity, endowing it as a feasible candidate for addressing electromagnetic pollution. Figure 1 schematically describes the fabrication process of CKF, where the collected KF (Figure 1b) was sequentially washed with water and ethanol to remove any impurities. Then, the air-dried KF was carbonized at 600 °C for 2 h under an argon atmosphere to obtain the CKF (Figure 1c), in which volatile constituents (such as CH_4_, CO_2_ and some organics) were removed and the intriguing porous architectures in KF were retained. The morphological features of the KF and CKF were visualized by SEM. It can be observed from Figure 1b′ that cylindrical KFs present regular hollow tubular structures (with a diameter of 8.0–10.0 μm and wall thickness of 0.8–1.0 μm), which is similar to Mohamed’s work [1]. It has a smooth surface due to the presence of inherent plant wax. Carbonization affords CKF a rough surface with subtle textures and wrinkles (Figure 1c’), indicating that the waxy coating has been removed from the fiber’s surface [35]. In addition, the diameter of the microtube was reduced from 15~30 μm for KF to 10~20 μm for CKF after carbonization.

To gain more insight into the variation of the nanostructure during carbonization, FTIR, TGA and XPS of the pre- and post-carbonized cylindrical samples were performed and the results are depicted in Figure 2. It can be clearly observed that there are various cellulose molecules with induced characteristic peaks, where peaks at 3372 cm^−1^, 2916 cm^−1^, 1738 cm^−1^ and 1248 cm^−1^ are ascribed to the stretching vibration of O–H, C–H, C=O and C–O, respectively [36]. The absorption peak at 1052 cm^−1^ is assigned to the carbohydrate or polysaccharide. Compared with those of KF, all these peaks suppress or disappear completely for CKF, implying that the carbon-containing compounds have converted into inorganic carbon. Figure 2b shows the thermogravimetric analysis (TGA) curve of KF, which simulates the temperature-dependent weight loss (%) during carbonization. Initially, the slight weight loss (∼5%) is observed over the temperature ranging from 30 to 85 °C, which is probably because of the evaporation of moisture content, including water or any other volatile compounds. Thereafter, the thermal decomposition of organic component begins at ~150 °C, then reaches its fastest rate at ~345 °C and completes at ~600 °C, evidenced by the dramatic weight loss from 95% to 25%. Finally, ~19% weight of the original sample remains, and this value stays the same even when the temperature further increases to 800 °C, indicating there are no extra chemical reactions at this stage. Although it has been confirmed that high specific area and porous carbon structures always form at a high carbonization temperature, 600 °C was selected for the carbonization of KF due to its high efficiency. The XPS survey spectra and high-resolution C1s spectra of the KF and CKF is shown in Figure 2c,d, respectively. The measured spectra show two peaks corresponding to C1s and O1s in the KF and CKF, respectively. The C1s spectrum ranging from 280.03 to 291.53 eV for KF can be further divided into three obvious peaks corresponding to C–C (284.75 eV), C–O (285.44 eV) and C=O (288.75 eV) groups, accounting for 45.07%, 49.86% and 5.07%, respectively. The increased C–C (63.59%) and decreased C–O (32.65%)/C=O (3.76%) indicate that KF has been converted into a carbon microtube during carbonization [32]. The amorphous structure of the resulting CKF can be further confirmed by its XRD pattern and Raman shift. As depicted in Figure 3a,b, there are two peaks at 15.9° and 22.1° corresponding to the (002) and (100) planes of CKF, respectively [37]. Evidenced by two peaks around 1338 cm^−1^ (D-band) and 1589 cm^−1^ (G-band) stemming from the stretching of in-plane C–C bonds and the defect of the carbon crystallite, amorphous carbon is formed in CKF. N_2_ adsorption–desorption experiment was performed to study the porosity of CKF. As shown in Figure 3c, the isotherm plots of N_2_ adsorption–desorption show a typical type-IV curve with a slight H4 hysteresis loop fast-rising over a wide range of P/P_0_, and the Brunauer–Emmett–Teller-specific surface area was measured to be 6.21 m^2^ g^−1^. The Barrett–Joyner–Halenda pore size distribution centered at approximately 1.48 and 6.34 nm demonstrates distinct micro- and mesoporous features of CKF.

SEM observations provide visual evidence for the morphology of CKF in paraffin. In the case of neat paraffin, the dense packed paraffin fully coalesces into a smooth surface due to the absence of CKF (Figure 4a). At low CKF concentration, only a few tubular CKFs as well as coalesced paraffin are observed composites (Figure 4b), indicating that such a low CKF loading is not sufficient to form an electromagnetic loss network. However, when the amount of CKF increases to 30 wt.% or even 40 wt.%, the CKF-based network becomes denser and results in the overlap of microtubes throughout the wax (Figure 4e,f). In other words, the more CKF is loaded, the more an interconnected electromagnetic loss network forms, which is in favor of an attenuating penetrated microwave.

### 3.2. Microwave-Absorbing Performance

Generally, the microwave-absorbing performance of an MAM is expressed by reflection loss (RL), and the smaller RL means less reflected energy. Normally, a RL value of lower than −10 dB refers more than 90% of penetrated electromagnetic energy can be effectively converted into another form of energy. Correspondingly, the frequency range in which RL value is smaller than −10 dB is defined as an effective absorption bandwidth [38], representing the frequency window for practical application. For a desired MAM, the RL should be as low as possible and the absorption bandwidth as wide as possible. On the basis of transmission line theory, the RL of an absorber can be calculated from its EM parameters according to the following formulas [2].
(1)$$RL=20\mathrm{lg}\left|\frac{Z_{in}-1}{Z_{in}+1}\right|$$
(2)$$Z_{in}=\sqrt{\frac{\mu_{r}}{\varepsilon_{r}}}\tanh\left(j\frac{2\pi fd}{c}\sqrt{\mu_{r}\varepsilon_{r}}\right)$$
where *Z_in_* is the normalized input impedance of the absorber, *ε_r_* is the complex permittivity, *μ_r_* is the complex permeability, *f* is the frequency of the microwave, *d* is the absorber thickness and *c* is the velocity of the microwave in free space. To evaluate the microwave absorption performance of obtained carbon microtube materials, the calculated *RL* is shown three-dimensionally in Figure 5a–e. It can be observed that the match frequency (*f_m_*) corresponding to minimum reflection loss (*RL*_min_) moves towards lower frequency when the absorber thickness increases, which can be explained by the quarter wavelength matching model in the following equation [38]:(3)tm=nλm/4=nc(4fm|εr||μr|12),n=1,3,5…

When the absorber thickness meets Equation (3), a phase cancellation effect occurs between the reflected microwave from the air absorber and absorber–backboard interface, resulting in the disappearance of the reflected waves.

It can be observed that CKF-5 and CKF-10 show an RL_min_ of less than −10 dB at the thickness range of 1.0–5.5 mm, indicating it is not of value in a practical application. The microwave-absorbing properties of CKF-filled paraffin composites become stronger when the loading of CKF increases from 5% to 30%, but deteriorates as the concentration increases further to 40%. Therefore, CKF-30 exhibits the best performance among all the samples, demonstrating a RL_min_ of −49.46 dB at the thickness of 2.3 mm and a broad EAB of 7.12 GHz (10.64–17.76 GHz, 2.3 mm) which covers 34% X-band and 96% of Ku-band is obtained. Moreover, more than 90% of incident electromagnetic waves over the frequency range of 4.48–18.00 GHz can be attenuated by CKF-30 by adjusting its thickness, which means CKF shows a feature of broadband absorption. Table 1 summarizes the typical reported biomass-derived MAMs in terms of loading, thickness, RL_min_ and EAB [20,21,24,25,39,40,41,42]. Studies indicate that most of the biomass-derived carbon absorbers are a form of hybrid with magnetic materials, which is compromised by complicated synthesis and high density. In this study, CKF demonstrates the balanced features of broader EAB, stronger absorption, lower filler loading and a thinner layer, making it highly comparable with its counterparts.

### 3.3. Microwave Attenuation Mechanisms

It is well known that the microwave-absorption performances of MAMs are associated with their relative complex permittivity (*ε*_r_ = *ε*′−*jε*″) and relative complex permeability (*μ*_r_ = *μ*′−*jμ*″), where the *ε*′ and *μ*′ stand for the capacity of energy storage, and the *ε*″ and *μ*″ represent the ability of energy attenuation [43]. To obtain more insights into the microwave attenuation mechanism of CKF, frequency-dependent electromagnetic parameters of the as-fabricated CKF–paraffin composites are plotted in Figure 6. It can be observed from Figure 6a,b that both the values of *ε*′ and *ε*″ for all composites decrease with the increase in frequency. This is in spite of a marginal decline for CKF-5 and CKF-10, which was ascribed to a polarization hysteresis resulting from a high-frequency electromagnetic field [44]. It is worth noting that the values of *ε*′ and *ε*″ gradually increase with the increase in CKF loading, indicating enhanced storage and dissipation capabilities. Increasement in *ε*′ and *ε*′ with the increased loading of CKF can be ascribed to a higher amount of mobile charge carriers, micro-capacitors and polarization centers at higher CKF concentrations, thus improving the microwave-absorbing performance of CKF-filled paraffin composites via conduction loss and polarization loss [19]. Owing to the absence of the magnetic loss component, the values of *μ*′ and *μ*″ for as-prepared CKF composites exhibit slight fluctuations around 1.0 and 0.0 (Figure 6a’,b’), implying negligible storage and dissipation capacity. Additionally, the frequency-dependent dissipation factors represented by the dielectric loss tangent (tan δ*_ε_* = *ε*″/*ε*′) and the magnetic loss tangent (tan δ*_μ_* = *μ*″/*μ*′) of all composites are calculated (Figure 6c,c’), where the dielectric loss-dominated attenuation mechanism can be further confirmed by the much higher values of tan δ*_ε_*.

Moreover, due to the extensive carbonization-induced defects in CKF and the abundant interfaces among CKF–paraffin composites, the corresponding orientation polarization and interfacial polarization also greatly contribute to the improvement of dielectric loss and favor the microwave-absorbing performance [45]. According to Debye theory, the Cole–Cole semicircle can be used to prove the presence of multiple dielectric relaxation, which can be deduced by the following equation [19]: (4)$${\left(\varepsilon^{\prime }-\frac{\varepsilon_{s}+\varepsilon_{\infty }}{2}\right)}^{2}+{\left(\varepsilon^{\prime \prime }\right)}^{2}={\left(\frac{\varepsilon_{s}-\varepsilon_{\infty }}{2}\right)}^{2}$$
where *ε*_s_ presents the static dielectric constant and *ε*_∞_ denotes the dielectric constant at the infinite frequency. As depicted in Figure 7, several distinguishable Cole–Cole semicircles are observed with the increase in CKF loading, implying the presence of multiple dielectric relaxation processes. With the increase in CKF loading, the semi-circles shift to a higher permittivity range and the radiuses increase simultaneously, indicating a higher polarization loss of CKF–paraffin composites with a higher CKF loading. In addition, the higher the CKF loading, the more heterointerface, charge transfer and defects, which further contribute to the enhancement of relaxation processes [20].

It has been widely accepted that the synergistic effects between magnetic loss and dielectric loss account for the final microwave-absorbing performance of an absorber, which can also be described by another two parameters. The first one is impedance matching (*Z*), which represents the ability of allowing as much of the incident EMWs entering into the absorber as possible. The other is the attenuation constant (*α*), which is associated with the capability of a dissipating penetrated wave in another form of energy. A desired MAM should allow as many electromagnetic waves to penetrate the MAM as possible, and then convert them into another form of energy. According to electromagnetic wave theory [2], *Z* and *α* can be evaluated by the following equations, respectively.
(5)$$Z=\left|\sqrt{\frac{\varepsilon_{r}}{\mu_{r}}}\right|$$
(6)$$\alpha =\frac{\sqrt{2}\pi f}{c}\sqrt{(\mu^{\prime \prime }\varepsilon^{\prime \prime }-\mu^{\prime }\varepsilon^{\prime })+\sqrt{{\left(\mu^{\prime \prime }\varepsilon^{\prime \prime }-\mu^{\prime }\varepsilon^{\prime }\right)}^{2}+{\left(\mu^{\prime }\varepsilon^{\prime \prime }+\mu^{\prime \prime }\varepsilon^{\prime }\right)}^{2}}}$$

Good impedance matching requires the values of *ε*_r_ and *μ*_r,_ to be close, so the values of *Z* should be as close to 1 as possible. Figure 8a depicts the calculated *Z* values for CKF-filled paraffin composites. It can be clearly seen that with the increase in CKF loading, the value of *Z* declines from 0.57~0.64 for CKF-5 to 0.49~0.59 for CKF-10, 0.37~0.54 for CKF-20, 0.30~0.45 for CKF-30 and 0.28~0.44 for CKF-40, respectively. Therefore, CKF-5 show the best impedance matching, confirmed by its *Z* value about 0.62. The calculated α curves for all as-prepared CKF samples are plotted in Figure 8b, where the values of α ascend with the increase in frequency, indicating that strong attenuation primarily occurs in a high frequency range. Further, it can be seen that the *α* values of CKF-filled composites elevates with the increase in CKF loading, which agrees with that of complex permittivity (Figure 6a–c), confirming that the microwave attenuation capacity of CKF is dominated by dielectric loss. Therefore, it is understood that the excellent microwave-absorbing performance of CKF-30 stems from its balanced impedance matching and attenuation capacity. Neither good impedance matching (CKF-5) nor strong attenuation ability (CKF-40) is enough to produce considerable RL characteristics [46].

According to the above analyses, a schematic diagram is proposed to illustrate the microwave attenuation mechanism of CKF composites (Figure 9). It can be summarized that the excellent microwave-absorbing performance of CKF primary stems from its proper concentration and intrinsic hierarchical structure, affording balanced impendence matching and attenuation capacity. For the former, moderate loading of CKF tunes the electromagnetic parameters of CKF-filled paraffin composites, thus improving impedance matching and decreasing the reflected wave. Additionally, carbonization-induced high surface and multiscale pores in CKF significantly regulate the permittivity, which is also beneficial to optimize the impedance matching condition. When it comes to the latter, multiple mechanisms including conductive loss derived from connected CKF networks, dipole polarization stemming from carbonization-induced defects, and interfacial polarization caused by abundant solid–void interface, contribute to the enhanced microwave attenuation capacity of CKF. Finally, high porosity and hierarchical structures of CKF generate multiple refection and scattering, which creates an extremely long transmission channel for the penetrated microwave, affording a greater chance for CKF to consume them.

## 4. Conclusions

In conclusion, a kapok-fiber-derived carbon microtube was fabricated by a versatile process, and a concentration-associated relation between morphology and performance has been systematically investigated. It has been found that carbonization-induced hierarchical structure and moderate concentration play crucial roles in the final microwave-absorbing performance. Due to the balanced impedance matching and attenuation capacity, CKF-30 with 30 wt.% CKF in paraffin demonstrate the best absorption performance, where the RL_min_ reaches −49.46 dB at 16.48 GHz for CKF-30 with a thickness of 2.3 mm. Moreover, an optimized effective absorption bandwidth of 7.12 GHz (from 10.64 GHz to 17.76 GHz) is obtained at the thickness of 2.3 mm. Therefore, it is evident that kapok-fiber-derived carbon microtube demonstrates great potential for applications in the field of broadband microwave attenuation.

## Figures and Tables

**Figure 1 materials-15-04845-f001:**
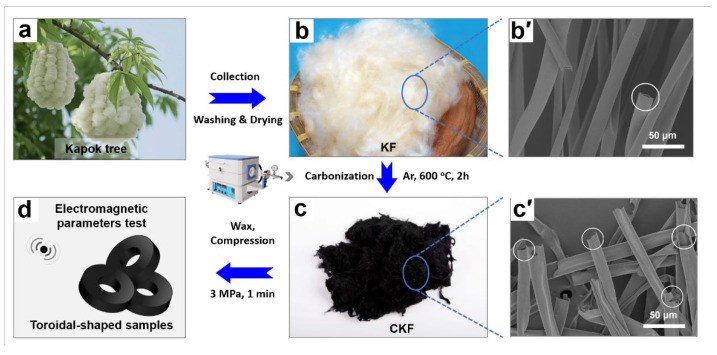
Schematic description of the fabrication of KF-derived carbon microtube for microwave attenuation.

**Figure 2 materials-15-04845-f002:**
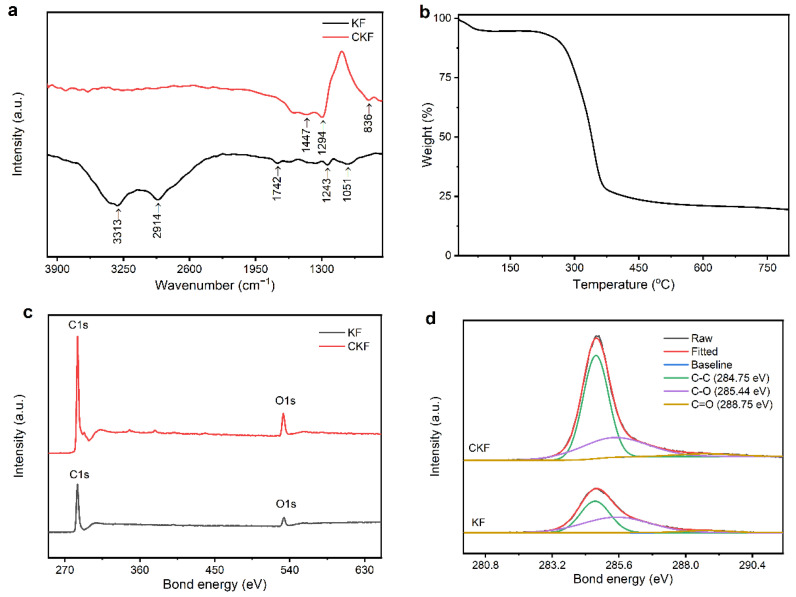
(**a**) FTIR of the KF and CKF; (**b**) TGA of the KF; (**c**) XPS survey spectra and (**d**) C1s spectra of the KF and CKF.

**Figure 3 materials-15-04845-f003:**
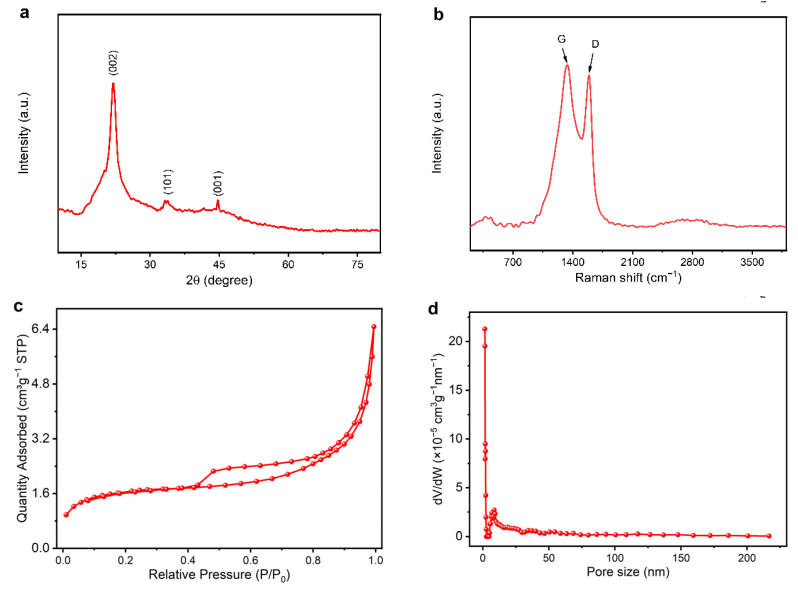
(**a**) XRD, (**b**) Raman, (**c**) nitrogen adsorption–desorption isotherm and (**d**) pore size distribution of CKF.

**Figure 4 materials-15-04845-f004:**
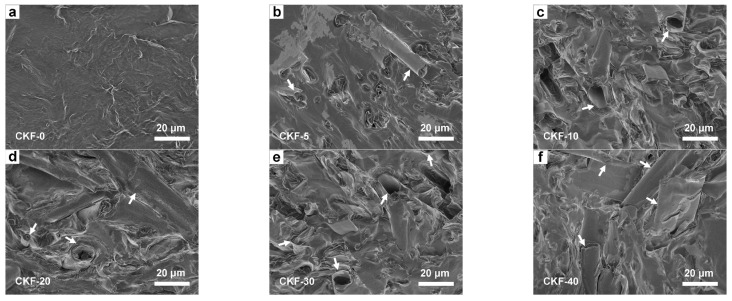
SEM of CKF-filled paraffin composites with different filler loading.

**Figure 5 materials-15-04845-f005:**
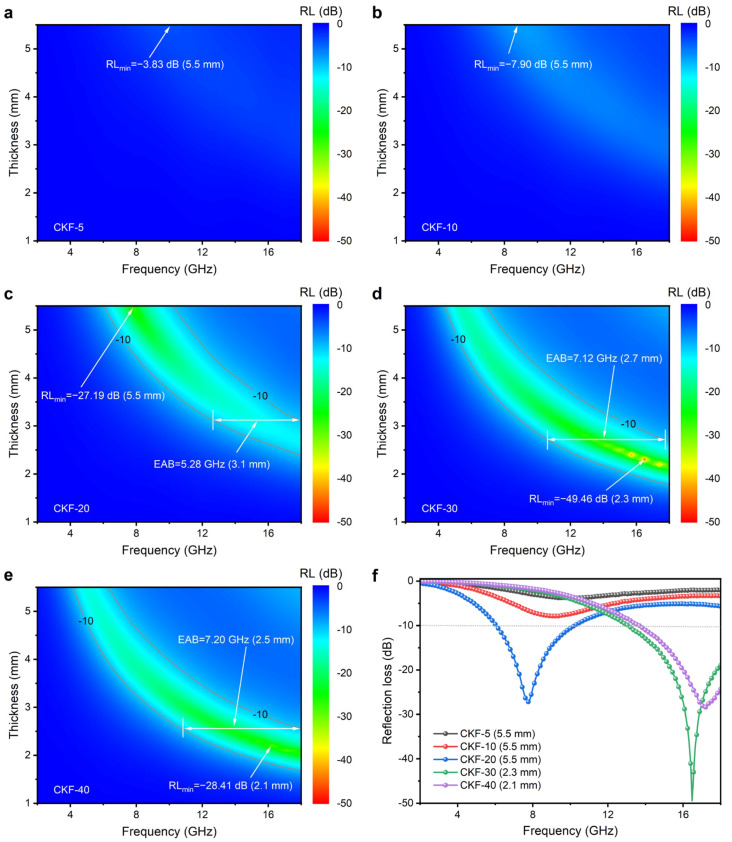
(**a**–**e**) Dependence of filler loading on the reflection loss and (**f**) extracted reflection loss curves at the optimized thickness of CKF-filled paraffin composites.

**Figure 6 materials-15-04845-f006:**
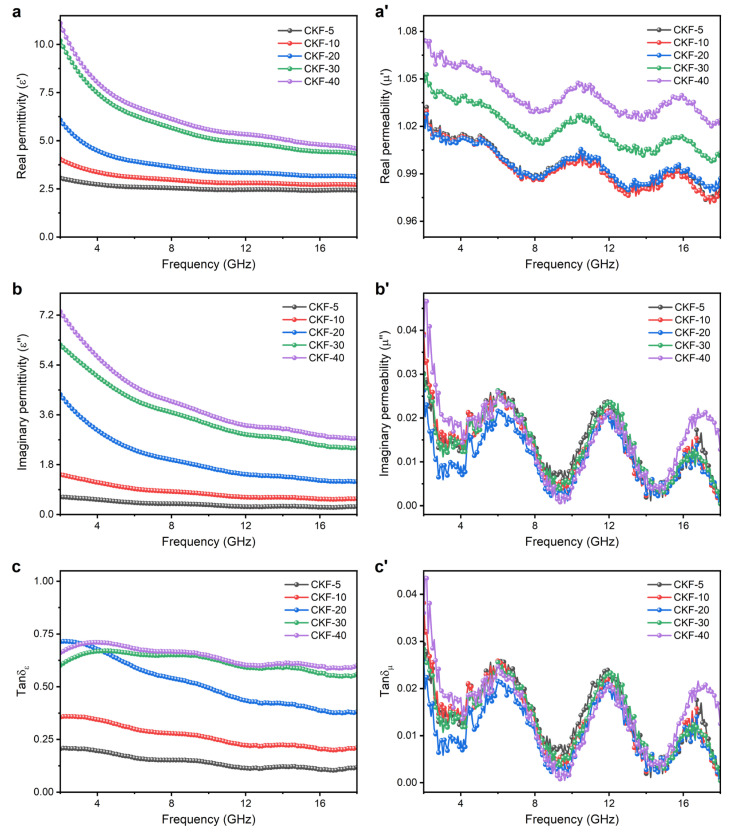
Frequency dependence of electromagnetic parameters of CKF-filled paraffin composites with different filler loading: (**a**–**c**) complex permittivity and (**a’**–**c’**) complex permeability.

**Figure 7 materials-15-04845-f007:**
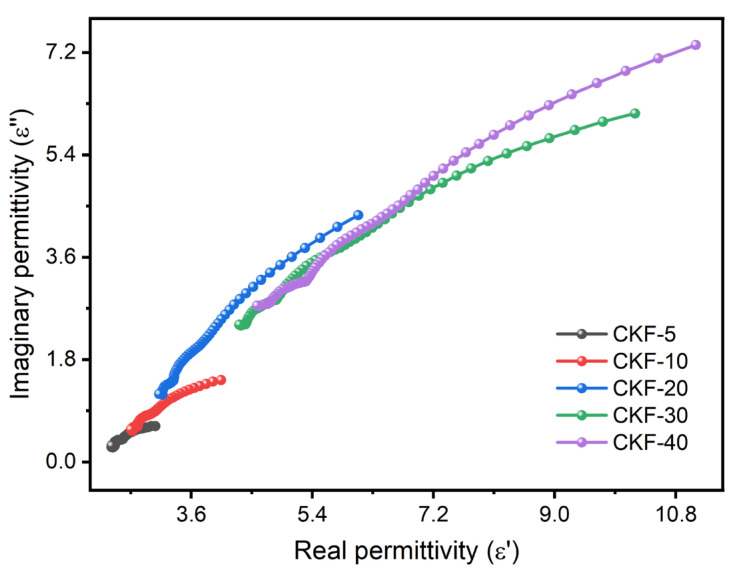
Cole–Cole curves of CKF-filled paraffin composites with different filler loading.

**Figure 8 materials-15-04845-f008:**
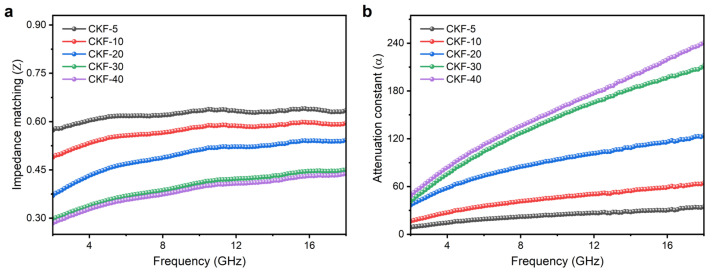
Calculated (**a**) impedance matching and (**b**) attenuation constant of CKF-filled paraffin composites with different filler loading.

**Figure 9 materials-15-04845-f009:**
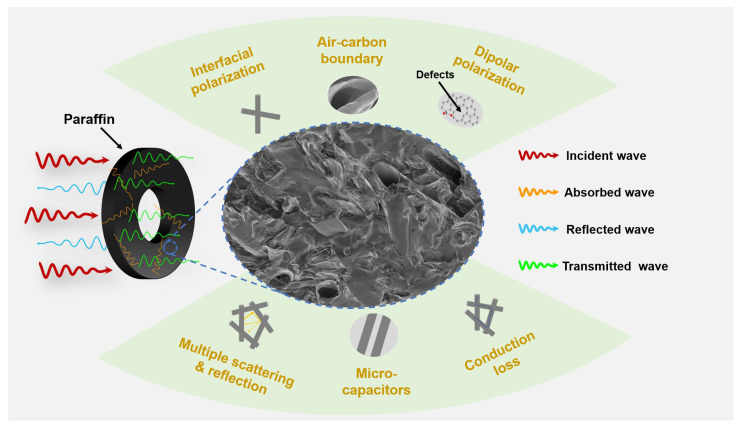
Schematic description of possible microwave attenuation mechanisms of 30 wt.% CKF-filled paraffin composites.

**Table 1 materials-15-04845-t001:** Typical microwave-absorbing materials based on biomass-derived carbon and their performances.

Biomass-Derived Microwave Absorber	Loading (wt.%)	Thickness (mm)	RL_min_ (dB)	EAB (GHz)	References
Walnut shell-derived porous carbon	70	2.00	−42.40	1.80 (8.08–9.84)	[20]
Mango-leaf-derived porous carbon	20	1.75	−23.60	5.17(12.83–18.00)	[21]
Rice-based porous C/Co	25	1.80	−40.10	2.70 (9.30–12.00)	[24]
Loofah-sponge-derived carbon/Fe_3_O_4_@Fe	30	2.00	−49.60	5.30 (13.00–18.00)	[25]
Porous carbon fiber/Fe_3_O_4_	30	1.90	−48.20	5.10(12.90–18.00)	[39]
NiO/porous carbon	30	8.00	−33.80	6.70 (11.30–18.00)	[41]
Porous carbon @NiFe_2_O_4_	30	2.50	−50.80	4.90 (12.40–17.30)	[42]
Kapok-fiber-derived porous carbon	30	2.30	−49.46	7.12 (10.64–17.76)	This work

## Data Availability

Not applicable.

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
