# Peer review of "Sustainable Kapok Fiber-Derived Carbon Microtube as Broadband Microwave Absorbing Material"

_materials, 2022, doi:10.3390/ma15144845_

Round 1

Reviewer 1 Report

Author presented a new KF based biochar material as EM absorbing material. The experimental results showed that the CKF materials showed an improved absorption bandwidth without forming a hybrid with magnetic materials.  The followings have to be addressed.

1. Authors need to explain more about the reason that they chose KF. Just stating that it has a tubular structure is not enough to be convinced.   

2. Authors stated that the CKFs are micro-tubular or hollow structure. However, reviewer cannot clearly see the structure of CKF from the figure 1 or figure 3 in the manuscript. Author needs to replace or substitute a figure to support the statement.

3.  Authors used toroidal shape paraffin/biochar samples. It was not clearly explained why the toroidal shape was used.    

4. In the caption of the figure 4 (f), it was stated that the RL was extracted from optimized thickness. However, the figure does not provide the data presented in the manuscript and is confusing. For example, the EAB of 7.28 GHz for CKF-30 could not be seen from figure 4 (f).

Author Response

Thank you very much for the comments concerning our manuscript entitled “Sustainable kapok fibers-derived carbon microtubes as broad-band microwave absorbing materials”. These comments are all valuable and very helpful for revising and improving our paper. We have checked the manuscript and made carefully corrections. We hope it meet the requirements for the publication of materials. All revisions to the manuscript were marked up using the “Track Changes” function, and our point-by-point responses to the comments are as following.

Responds to comments:

  1. Authors need to explain more about the reason that they chose KF. Just stating that it has a tubular structure is not enough to be convinced. 

Response: There are three reasons to choose KF. Initially, KF is sustainable and easily accessible, affording a green and cost-effective source of KF-based microwave absorbing materials (MAMs). Then, the presence of microscale pores in KF not only facilities impedance matching via abundant air-solid interfaces, but also contribute to microwave attenuation via multiple reflections. Moreover, KF is easy to be processed and modified, making it is versatile to develop KF based hybrids for microwave attenuation. This work is just the beginning, many strategies including doping with other atoms, hybridizing with nano-blocks and compounding with polymer are carrying in our lad, which will facilities KF as one of the most promising MAMs in the future.

  1. Authors stated that the CKFs are micro-tubular or hollow structure. However, reviewer cannot clearly see the structure of CKF from the Figure 1 or Figure 3 in the manuscript. Author needs to replace or substitute a figure to support the statement.

Response: Figure 1 and Figure 3 have been improved for better illustration. Specific modifications can be found on page 3, line 130 and page 5, line 194.

  1. Authors used toroidal shape paraffin/biochar samples. It was not clearly explained why the toroidal shape was used.

Response: There are three methods can be adopted to characterize the microwave absorbing performance of an absorber. While waveguide method only provides electromagnetic parameters in limited frequency, arc method only provides the reflection loss of an absorber with given thickness. As coaxial method is able to provide electromagnetic parameters in wide frequency, it is widely used to measure the microwave absorbing performance. Therefore, toroidal-shaped samples were used to characterize the electromagnetic parameters, thus deducing the reflection loss at different thickness.

  1. In the caption of the figure 4 (f), it was stated that the RL was extracted from optimized thickness. However, the figure does not provide the data presented in the manuscript and is confusing. For example, the EAB of 7.28 GHz for CKF-30 could not be seen from figure 4 (f).

Response: Generally, the RLmin and EAB always occur at different thicknesses. Figure 4f in original manuscript demonstrated the RL curves at the thickness for RLmin, not the RL curves at the thickness for EAB. For better comparison, RLs of the CKF samples are plotted in contours. Specific modifications can be found on page 7, line 233.

Again, thank you very much for the comments and suggestions. If there are any problems or questions about our manuscript, please feel free to contact us.

Yours sincerely,

Pengfei Zhao

Associate Professor in Material Science and Engineering

Key Laboratory of Tropical Crop Products Processing of Ministry of Agriculture and Rural Affairs, Agricultural Products Processing Research Institute, Chinese Academy of Tropical Agricultural Sciences, Zhanjiang 524001, China

E-mail: pengfeizhao85ac@163.com

Tel: +86 18666729539

Reviewer 2 Report

In this research, carbon microtubes were fabricated using kapok fibers as biomass through a facile pyrolysis route. The architected structure was characterized by X-ray diffraction (XRD), scanning electron microscopy (SEM), Fourier 103 transformation infrared spectrophotometer (FTIR), thermogravimetric analysis (TGA), Raman, and X-ray photoelectron spectroscopy (XPS) analyses. Eventually, the microwave absorbing features of the tailored samples were evaluated using a paraffin matrix with diverse mass fractions of fillers where CKF-30 gained a maximum reflection loss of -49.46 dB at 16.48 GHz with a thickness of 2.3 mm and efficient bandwidth of 7.28 GHz at 2.7 mm in thickness.

There are several points that should be considered by the authors:

1.                 Why the mentioned range of the analyses is different in “2.3. Characterizations” and “3. Results”?, e.g. FTIR and XRD.

2.                 Please further untangle the introduction based on cutting-edge bio-inspired materials used as microwave absorbing components. Diverse carbon-based structures have been reduced using the thermal pyrolysis and unique morphologies of this type of material have been tailored. Recent researches have clarified that the remained heteroatoms, morphology, phase, and defects pave the way for the microwave absorbing capability of these types of materials (DOI: 10.1016/j.carbon.2022.02.015; 10.1007/s40820-021-00635-1).

3.                 What are the reasons behind the observed fluctuations in the permeability curves? The observed permeability in non-magnetic structures is interesting (DOI= 10.1016/j.ceramint.2022.03.314; ).

4.                 The author has discussed porosity as an essential factor paving the way for microwave absorbing capability. It is suggested that N2 adsorption/desorption analysis be provided.

5.                 Please revise the phrases based on the organic chemistry definition. “Compared with KF, all these peaks suppress or disappear completely for CKF, implying that the organic carbon has converted into inorganic counterpart.”; “The increased C−C (63.59%) and decreased C−O (32.65%)/C=O (3.76%) indicate the removal process of organics during carbonization”

6.                 Please revise the manuscript carefully, and correct some wrong grammars and spelling, e.g. revise Lines 251 and 252: “Increase in ε′ and by ε′”.

Author Response

Thank you very much for the comments concerning our manuscript entitled “Sustainable kapok fibers-derived carbon microtubes as broad-band microwave absorbing materials”. These comments are all valuable and very helpful for revising and improving our paper. We have checked the manuscript and made carefully corrections. We hope it meet the requirements for the publication of materials. All revisions to the manuscript were marked up using the “Track Changes” function, and our point-by-point responses to the comments are as following.

Responds to comments:

  1. Why the mentioned range of the analyses is different in “2.3. Characterizations” and “3. Results”? e.g., FTIR and XRD.

Response: As there is no obvious peaks, the original figures only show the data in limited range. For the sake of rigorous, figures covering all range are provided in revised manuscript. Please see specific modification on page 3, line 118 and page 5, line 194.

  1. Please further untangle the introduction based on cutting-edge bio-inspired materials used as microwave absorbing components. Diverse carbon-based structures have been reduced using the thermal pyrolysis and unique morphologies of this type of material have been tailored. Recent researches have clarified that the remained heteroatoms, morphology, phase, and defects pave the way for the microwave absorbing capability of these types of materials (DOI: 10.1016/j.carbon.2022.02.015; 10.1007/s40820-021-00635-1).

Response: It is well-accepted that the introduction of magnetic lossy component will boost the microwave absorbing performance, lanthanide metal oxide anchored CKF is being carried in our lab and some intreseting results have been obtained. The provided artciles are of great values to improve our work, which will be cited in our follwing artile as it is out of the span of current work.

  1. What are the reasons behind the observed fluctuations in the permeability curves? The observed permeability in non-magnetic structures is interesting (DOI= 10.1016/j.ceramint.2022.03.314;).

Response: As similar to the fluctuations in the permittivity curves, slight fluctuations in the permeability curves may result from resonance effect in alternating electric field. Herein, owing to the absence of magnetic loss component, the values of μ′ and μ″ for as-prepared CKF composites exhibit slight fluctuations around 1.0 and 0.0 (Figure 6a′ and 6b′), implying their negligible storage and dissipation capacity. Specific modifications can be found on page 8, line 272.

  1. The author has discussed porosity as an essential factor paving the way for microwave absorbing capability. It is suggested that N2 adsorption/desorption analysis be provided.

Response: BET result has been provided and discussed in revised manuscript. Detailed information can be found on page 5, line 188.

  1. Please revise the phrases based on the organic chemistry definition. “Compared with KF, all these peaks suppress or disappear completely for CKF, implying that the organic carbon has converted into inorganic counterpart.”; “The increased C−C (63.59%) and decreased C−O (32.65%)/C=O (3.76%) indicate the removal process of organics during carbonization”

Response: Thanks. All similar questions have been addressed.

  1. Please revise the manuscript carefully, and correct some wrong grammars and spelling, e.g., revise Lines 251 and 252: Increase in ε′ and by ε′”.

Response: The manuscript had been carefully checked, special attention was paid to grammar, spelling, sentence structure. Before submission, the manuscript has been refined by our professional colleague from Australia.

Again, thank you very much for the comments and suggestions. If there are any problems or questions about our manuscript, please feel free to contact us.

Yours sincerely,

Pengfei Zhao

Associate Professor in Material Science and Engineering

Key Laboratory of Tropical Crop Products Processing of Ministry of Agriculture and Rural Affairs, Agricultural Products Processing Research Institute, Chinese Academy of Tropical Agricultural Sciences, Zhanjiang 524001, China

E-mail: pengfeizhao85ac@163.com

Tel: +86 18666729539

Round 2

Reviewer 2 Report

Accepted.